# The Fragmented Picture of Antimicrobial Resistance in Kenya: A Situational Analysis of Antimicrobial Consumption and the Imperative for Antimicrobial Stewardship

**DOI:** 10.3390/antibiotics13030197

**Published:** 2024-02-20

**Authors:** Aarman Sohaili, Judith Asin, Pierre P. M. Thomas

**Affiliations:** 1Faculty of Health, Medicine and Life Sciences, Maastricht University, 6229 ER Maastricht, The Netherlands; aarman.sohaili@outlook.com; 2Pharmaceutical Systems Strengthening Lead, Ecumenical Pharmaceutical Network, P.O. Box 749, Nairobi 00606, Kenya; 3Institute of Public Health Genomics, Genetics and Cell Biology Cluster, GROW Research School for Oncology and Development Biology, Maastricht University, 6229 ER Maastricht, The Netherlands

**Keywords:** antimicrobial resistance (AMR), Kenya, situational analysis, policy implementation

## Abstract

Antimicrobial resistance (AMR) jeopardizes the effectiveness of essential antimicrobial agents in treating infectious diseases. Accelerated by human activities, AMR is prevalent in Sub-Saharan Africa, including Kenya, due to indiscriminate antibiotic use and limited diagnostics. This study aimed to assess Kenya’s AMR efforts through a situational analysis of policy efficacy, interventions, and implementation, culminating in recommendations for strengthening mitigation. Employing two methodologies, this study evaluated Kenya’s AMR endeavors. A systematic scoping review summarized AMR dynamic, and an expert validated the findings, providing an on-the-ground perspective. Antibiotic resistance is driven by factors including widespread misuse in human medicine due to irrational practices, consumer demand, and substandard antibiotics. Heavy antibiotic use in the agricultural sector leads to contamination of the food chain. The National Action Plan (NAP) reflects a One Health approach, yet decentralized healthcare and funding gaps hinder its execution. Although AMR surveillance includes multiple facets, diagnostic deficiencies persist. Expert insights recognize proactive NAP but underscore implementation obstacles. Kenya grapples with escalating resistance, but commendable policy efforts exist. However, fragmented implementations and complexities persist. Addressing this global threat demands investment in healthcare infrastructure, diagnostics, international partnerships, and sustainable strategies.

## 1. Introduction

Antimicrobial agents, such as antibiotics, antivirals, antifungals, and antiparasitics, play a crucial role in preventing and treating infectious diseases. However, the emergence of antimicrobial resistance among pathogens poses a significant threat to public health worldwide [1]. In a 1945 interview, Sir Alexander Fleming, renowned for his discovery of the first highly effective antibiotic, penicillin, forewarned about the potential emergence of resistant bacteria resulting from excessive antibiotic use, with this prediction now materializing [2]. AMR is increasingly recognized as a major global health and economic burden, with far-reaching effects on mortality, morbidity, healthcare costs, and productivity [3]. Economically, AMR has been projected to decrease the global gross domestic product by approximately 1% by 2050, in addition to causing the loss of millions of lives if the current AMR trends continue [4].

AMR occurs naturally over time as microbes adapt and develop genetic changes; however, human activities, such as overuse and misuse of antimicrobial agents and inadequate infection prevention measures, have expedited this process [4]. Resistant bacteria can spread globally through cross-reservoir transmission, making the One Health approach essential for addressing AMR [5]. The One Health approach acknowledges collaborative efforts among various disciplines to tackle issues concerning human, animal, and environmental health. Recognizing these connections, One Health mobilizes various disciplines and communities across various sectors to collaborate and promote health by tackling AMR-related challenges. In response to the gravity of the situation, the World Health Organization (WHO) developed a Global Action Plan (GAP) for AMR in 2015. The plan emphasizes a collaborative One Health approach, urging member states to develop a NAP aligned with the WHO’s strategies.

These strategies include improving awareness, enhancing surveillance, implementing effective infection prevention measures, reducing the use of antibiotics for human and animal health, and increasing investment in research and development [6]. To enhance surveillance efforts, the WHO established the Global Antimicrobial Resistance and Use Surveillance System (GLASS) in 2014. GLASS advocates for a holistic approach to surveillance, incorporating epidemiological, clinical, and population-level data, to provide a standardized framework that enables countries to collect analyze, interpret, and share data [7]. 

AMR is a global challenge affecting all countries regardless of their socioeconomic status [8,9]. However, in Africa, the problem of AMR is exacerbated by several factors, including indiscriminate antibiotic use, poor sanitary conditions, inadequate healthcare systems with limited diagnostic capabilities, and a lack of access to quality antibiotics [10,11,12]. Kenya, situated in East Africa, faces significant challenges, with over 43% of its population living in poverty. The country grapples with pressing health issues, including elevated rates of maternal and child mortality, along with a substantial burden of infectious diseases such as HIV, tuberculosis and malaria [13]. Consequently, there is concern regarding the rise in AMR to commonly used first-line drugs and increasing infections from life-threatening pathogens [13,14].

Alarmingly, carbapenem-resistant Enterobacterales (CRE) and extended-spectrum-lactamase-producing organisms, which are associated with poor clinical outcomes in invasive infections, are increasingly prevalent in Kenya. A recent study found that 7–17% of hospitalized patients had infections caused by these bacteria, with 39% of isolates meeting the “difficult-to-treat resistance” definition [15]. This situation poses a significant public health challenge requiring urgent attention. Compounding this is the lack of routine surveillance data and methodological limitations that severely constrain the applicability of local data on antibiotic usage and resistance [16]. Kenya has actively engaged in AMR surveillance since the 1980s; however, the lack of microbiology laboratories in most healthcare facilities hinders the accurate measurement of AMR [16].

Despite the abundance of policies addressing AMR in Kenya, there is a noticeable lack of studies assessing policy implementation progress and the necessary actions for swift execution. This paints a complex and fragmented picture of the AMR landscape in Kenya. To address this gap, this study aimed to conduct a systematic situational analysis encompassing policy effectiveness and the implementation of AMR interventions. Furthermore, insights from an expert in the field will offer an on-the-ground evaluation of ongoing initiatives. We envision that researchers can use our paper to navigate the diverse responsibilities associated with AMR, particularly in LMICs, ensuring that their data have a substantial impact. While we acknowledge the existing uncertainties in the landscape, particularly in the global south, our manuscript is crafted to initiate a meaningful discussion and set a trajectory for future research. The objective is to contribute valuable insights to the ongoing dialogue on resolving drug resistance issues in the global south, where vulnerability to such challenges is pronounced. Our emphasis lies in establishing the groundwork for continued exploration and understanding, aiming to foster a holistic approach to addressing AMR.

## 2. Results

The search process identified potential articles that were screened for language and duplicates based on titles and abstracts. After the initial screening, potentially relevant papers were selected for further review. Subsequently, full-text screening was conducted by applying the inclusion criteria, leading to the exclusion of articles. During the data extraction stage, articles were excluded if they did not mention an AMR variable of interest or if their geographic location did not include Kenya. This thorough process resulted in the final set of articles (*n* = 19) being included in the review, as depicted in Figure 1.

Among the included studies, regarding AMR variables of interest, most studies focused on an AMR situational analysis (*n* = 9). Three studies analyzed AMR surveillance and diagnostics, as well as Antimicrobial Stewardship (AMS). Two studies concentrated on AMR from a One Health perspective and an AMR communications strategy, respectively. Detailed information regarding the characteristics of all included studies is shown in Table 1.

## 3. Drivers of Antibiotic Resistance

### 3.1. Socio-Economic Factors and Related Behaviors

Rapid socioeconomic development in Kenya has not only enhanced the overall quality of life but also exacerbated wealth inequity among differing regions and demographic groups. Beyond its direct influence on healthcare quality and availability, SES profoundly impacts an individual’s lifestyle and behavior, including the use of antibiotics. A positive association has been shown between factors such as warmer temperatures, poorer administrative governance, higher urbanization levels, and a higher ratio of private to public health expenditure with the development and spread of AMR [31]. In the context of Kenya, these characteristics are evident as the nation faces the challenges of limited resources, sanitation and food safety issues, lenient regulations, high population density, and socioeconomic disparities. The convergence of these factors creates an environment conducive to detrimental behaviors and practices related to antibiotic use [2].

### 3.2. Antibiotic Consumption

#### 3.2.1. Antibiotic Use in Human Medicine

A combination of behavioral factors and economic incentives drives inappropriate prescription, dispensing, and purchasing of antibiotics. In Kenya, key drivers of AMR were highlighted in the overuse of antibiotics in the medical, veterinary, and agricultural fields [23]. Further evidence from a study in KNH CCUs showed that only 18.5% of antibiotic usage demonstrated a rational approach, with common irrational practices including the inappropriate selection of antibiotics (51%) and incorrect duration of treatment (32.3%) [29]. Consumer demand for antibiotics is fueled by a desire for rapid symptom eradication. Informal pharmacies, consisting of approximately 66% of the estimated 12,000 private pharmacies in Kenya, vastly outnumber certified branches [32]. 

Although laws and regulations specify that antimicrobial agents should only be dispensed, sold, and used with prescriptions from licensed clinicians or animal health professionals [33], antimicrobials are readily available over the counter or via unregulated supply chains [13]. Overall patient encounters with antimicrobial agents in East Africa was 57%, surpassing the WHO recommended value of ≤20% [27]. Another institutional factor contributing to the rise of AMR in Kenya is the influx of counterfeit antibiotics into the global pharmaceutical market [5]. Counterfeit products, often originating from Southeast Asia and Africa, are widely circulated. In Kenya, the quality of antibiotics on the market is largely unknown, and a significant proportion (up to 30%) could fail tests for labeled potency [16]. For instance, amoxicillin samples from different brands sourced from pharmacies in Nairobi were examined, and it was found that a significant 37.7% of these samples did not comply with the standards set by the United States Pharmacopoeia [34].

Adding to the fragmented AMR landscape in Kenya is the distinct disparity in healthcare service delivery. The private for-profit healthcare sector, represented by entities like Aga Khan Healthcare, MP Shah, and African Air Rescue, has expanded significantly, especially in urban areas, offering flexibility and efficiency. In addition to the wide array of operating CSOs and private non-profit organizations, despite their diverse services attracting lower-income individuals, they constitute a disjointed system [35]. Challenges include higher healthcare costs, limited health literacy among patients, inadequate self-management support, and ineffective referral systems [36]. 

Furthermore, the lack of vertical integration between healthcare levels impedes coordination, worsening AMR management in public and private facilities. Disjointed communication and skewed interactions among actors, coupled with limited government oversight, exacerbate the issue.

#### 3.2.2. Antibiotic Use in Animal Health and Agriculture

The increasing demand for animal protein, primarily in Asia and Africa, has led to the routine use of antimicrobials in veterinary practice and livestock production [37]. In Kenya, antibiotics are commonly used for therapeutic and prophylactic purposes, with therapeutic applications accounting for approximately 90% of all antibiotic purchases. Unfortunately, a multitude of farmers rely on antibiotics to treat animal illnesses rather than implementing proper hygiene and feeding practices [16]. Moreover, the Kenya Veterinary Association estimates that 33% of antibiotics available for animal use are substandard and/or counterfeit, with over 78% of veterinary medicine outlets operating illegally throughout the country [16]. This misuse of antibiotics has the potential to lead to the accumulation of antibiotics in animal products sold for human consumption, thereby contributing to the spread of antibiotic-resistant bacteria through the food chain. 

For instance, backyard poultry and cattle farming are essential to household economies in rural areas. Antibiotic use in poultry is particularly high, accounting for nearly 20% of the mean consumption per year [16,38]. This is primarily through the use of antimicrobial growth promoters (AGPs), which are believed to prevent disease, optimize feed conversion, promote growth and improve gut health [39]. However, excessive use of antibiotics in animal husbandry and close proximity to animals can lead to direct zoonotic transmission, with additional environmental contamination occurring through farm manure-contaminated water [40].

### 3.3. Antibiotic Resistance

#### 3.3.1. AMR Surveillance

To enhance the availability of AMR data through surveillance and research, Kenya has developed a national strategy complementing the NAP on AMR. The strategy aimed to strengthen data collection, promote routine AST, establish national databases and biobanks, monitor AMR trends, and inform clinical treatment guidelines [41]. The Kenya Medical Research Institute primarily leads AMR data generation, supported by central reference laboratories, national hospitals, and sentinel sites focusing on eight priority pathogens aligned with the WHO [13]. As of 2021, Kenya has ten facilities, including five surveillance sites and five healthcare facilities, participating in the national surveillance system reporting to GLASS. These facilities rely on clinical isolates collected from tertiary-level surveillance facilities. In the animal health sector, six laboratories have contributed data to the national database, with plans to integrate human and animal health databases in the future [13].

#### 3.3.2. AMR Laboratory and Diagnostic Capacity

In 2021, the MOH issued guidelines stressing the importance of performing antimicrobial susceptibility testing (AST) before initiating antibiotic treatment to curb unnecessary prescriptions [13]. Unfortunately, limited access to microbiological services, particularly in severe cases, leads to widespread administration of broad-spectrum antibiotics, as obtaining culture results within a timely intervention window proves challenging [16]. The inadequacy of laboratory infrastructure, especially in rural areas where basic requirements are rarely met, poses a significant barrier to reliable pathogen detection and AST [8]. Moreover, there exists a persistent healthcare delivery imbalance across counties, with over twice the laboratory technology accessibility in urban areas compared to their rural counterparts in Kenya [19].

Furthermore, challenges persist in AMR data management due to understaffing, the absence of professional standards, and insufficient training for clinical and laboratory staff. This exacerbates issues like shortages in essential items, the use of low-quality diagnostics, and the lack of local manufacturing capabilities, all of which compromise the accuracy of laboratory results. Additionally, the absence of clear guidance for specimen selection, transportation, and quality assurance, coupled with difficulties in updating procedures due to language and cultural barriers, further hinders progress [42]. The integration of internationally standardized criteria for bacteria, as advocated by organizations like the Clinical and Laboratory Standards Institute, remains lacking in low- and middle-income countries such as Kenya [8].

### 3.4. Antimicrobial Stewardship

#### 3.4.1. AMS and the One Health Approach

Kenya’s One Health approach gained prominence during the Rift Valley Fever Outbreak of 2006–2007. This crisis fostered collaboration among international organizations, researchers, and government entities. As a result, the adoption of the One Health approach in Kenya has enabled swift detection and effective control of zoonotic disease outbreaks at their source [43]. Integral to Kenya’s AMR strategy is its NAP on AMR (Figure 2). Developed in 2017, the NAP aimed to enhance healthcare quality, mitigate the economic ramifications of AMR, and align with the World Health Organization’s Global Action Plan on AMR [41].

Rooted in evidence-based recommendations from situational analyses conducted in 2011 and 2016, the NAP embraces a holistic approach by integrating One Health principles [13]. Acknowledging Kenya’s commitment to the One Health concept, the Global One Health (GOHI) index was utilized; this comprehensive tool assesses and ranks nations based on their efforts to address global AMR trends. Kenya received a high-ranking value of 41.30–52.63, surpassing the average ranking of 32.99 for LMICs. This positioning showcases Kenya as a prominent example of One Health advocacy within the region, underscoring the nation’s dedication to addressing AMR through a collaborative and integrated approach [44].

**Figure 2 antibiotics-13-00197-f002:**
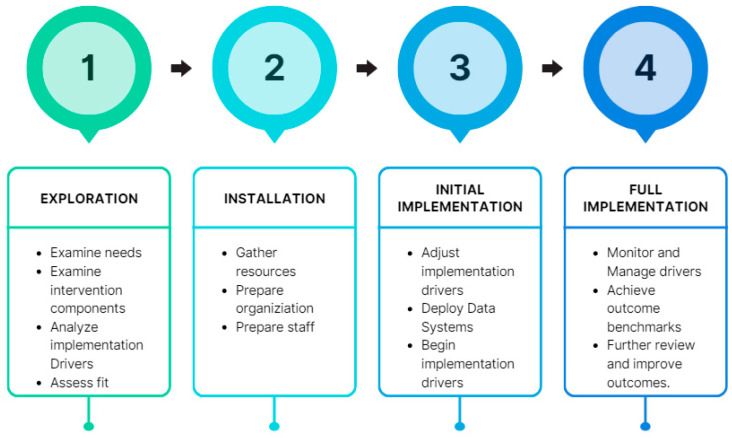
Implementation stages of the NAP and follow-up in Kenya [24,45].

The NAP on AMR assigns crucial responsibilities to various stakeholders, including national and county governments. To facilitate effective implementation, a well-defined hierarchy is in place (Figure 3), which includes a two-tiered coordination system established at both the national and county levels:At the national level, the National Antimicrobial Stewardship Interagency Committee (NASIC) and Technical Committees oversee and guide AMR-related activities. These committees report to the MOH and the Ministry of Agriculture, Livestock, Fisheries, and Cooperatives for guidance and funding.At the county level, 8 of Kenya’s 47 counties have County Antimicrobial Stewardship Interagency Committees (CASICs). These committees play a pivotal role in monitoring NAP implementation and allocating resources for AMR-related efforts [13].

Despite considerable progress, several challenges hinder the full realization of the NAP’s objectives [9]. The NAP’s update, originally scheduled for 2022, has been delayed owing to the unexpected emergence of the COVID-19 pandemic. This crisis disrupted AMR-related activities across the nation, diverting resources and attention from AMR mitigation efforts [18]. Notably, Kenya’s decentralized health system has posed challenges, with numerous counties not actively engaging in AMR mitigation efforts [33].

#### 3.4.2. AMR Awareness and Communication

The question “does AMR have a face?” arose in a sensitization session for journalists in Kenya, revealing the challenge of visualizing AMR as a distinct condition compared to diseases like TB and HIV/AIDS [25]. Despite its complexity, the Kenyan government aims to raise awareness through campaigns to promote understanding among stakeholders. In Nairobi’s Kibera settlement, a study revealed widespread misconceptions about antimicrobials, with 66% mistakenly believing they could treat influenza and the common cold. Less than half received proper information on antibiotic use, recognizing clinicians and pharmacists as trusted sources [20]. Efforts to address limited medical training include integrating AMR education into healthcare professionals’ curriculums. Despite initiatives, a lack of awareness persists, as outlined in Kenya’s NAP on AMR. A national communication strategy aimed to support AMR awareness campaigns but faced funding challenges like a myriad of other state health initiatives [46].

Another factor that contributes to the ambiguity surrounding AMR in Kenya is the notable absence of equity and local representation in AMR research. Although AMR research is on the rise in Kenya and other LMICs, the trend in authorship disparity persists. Studies focusing on African infectious disease research have revealed persistent imbalances, with the majority of lead authors coming from high-income countries [47]. Collaborations between high- and low-income institutions in international health research often lead to imbalances because high-income partners tend to drive primary research objectives and provide funding. Consequently, low-income-country-affiliated authors face barriers such as limited promotion and leadership opportunities, less emphasis on local health priorities, and restricted access to academic and medical literature [48].

#### 3.4.3. IPC, Water, Sanitation and Hygiene (WASH) and Immunization

The convergence of IPC, WASH, and immunization efforts represents a multi-faceted strategy to combat the escalating challenge of AMR [49]. The establishment of a national IPC program, complete with guidelines, reflects the country’s commitment to curtailing infections and subsequent antibiotic demand. However, the intricate link between WASH and IPC cannot be overlooked, as suboptimal WASH infrastructure, particularly limited access to clean water and sanitation facilities, poses a significant obstacle to effective control measures [13]. This deficiency potentially exacerbates AMR concerns by creating environments that are conducive to the proliferation of resistant pathogens.

Kenya’s impressive national vaccination rate is pivotal for the reduction in infections that require antibiotic treatment. The imminent transition toward self-funded immunization underscores the need for ongoing efforts to ensure equitable access to vaccines, thereby perpetuating lowered infection rates and consequently diminishing the overall burden of antibiotic consumption [13].

#### 3.4.4. Funding for Programs

Concurrently, a recurring theme in AMR interventions is the substantial dependence on external funding from institutions such as the Fleming Fund, WHO, and USAID, which are pivotal in bolstering AMS initiatives via training, material support, and laboratory equipment procurement [13,26]. These initiatives are well aligned with WHO guidelines and include capacity-building, procurement of necessary equipment, and mentorship. However, a persistent challenge faced by AMR-focused programs in Kenya is their heavy reliance on foreign aid [50]. Scrutinized over the past decade for its perceived failure to achieve desired outcomes, the effectiveness of foreign aid in promoting growth is heavily contingent on a favorable policy environment. 

This situation can be likened to a double-edged sword: while aid can be highly beneficial in supporting progress when policy and economic conditions are conducive, it can be ineffective or even counterproductive when such conditions are lacking [50]. At its worst, aid can inadvertently sustain corrupt or incompetent governments, as the reliance on foreign intervention nurtures a mentality of “let the West handle it” [51].

## 4. Expert Opinion

Exploring Kenya’s AMR initiatives requires understanding expert perspectives on obstacles and pathways for progress. Insights from this interview highlight current challenges and strategies shaping Kenya’s regulatory landscape and on-the-ground situation. Kenya’s pursuit to combat AMR is underpinned by a multifaceted strategy. The nation’s proactive measures, as seen through the development of an extensive NAP on AMR, set it apart from the other countries in the region. Notably, the NAP integrates NASIC and CASIC to ensure a coordinated country-wide approach. However, challenges persist in spurring the NAP into action, particularly in humble healthcare facilities with limited resources and awareness. 

This gap stems from an imbalance in the focus on public health facilities, creating hurdles for consistent NAP implementation across all healthcare settings. While urban facilities benefit from better-trained healthcare professionals, rural areas often grapple with a lack of expertise, even extending to nurses dispensing medications due to the lack of urban–rural migration. To address gaps, Kenya embraced collaboration with civil society organizations (CSOs) and faith-based groups. These partnerships aim for inclusivity, fostering discussions among CSOs, the government’s AMR focal person, and diverse healthcare professionals. The strategy focuses on aligning initiatives with the NAP framework. 

Central to this approach is establishing AMS programs at health facilities, recognizing them as key service delivery sites. AMR stewardship committees play a crucial role in overseeing antibiotic use, improving governance, and ensuring IPC practices. Proactively, CSOs and faith-based organizations designate AMR and IPC champions, providing specialized training to develop action plans for enhancing these aspects within their facilities. Nevertheless, challenges in laboratory capacity hinder data collection on AMR trends.

Limited financial resources for reagents and equipment, coupled with a reluctance towards testing practices among healthcare professionals, impede diagnostics. Additionally, addressing substandard medications has emerged as a crucial yet overlooked issue. Faith-based organizations are employing innovative solutions, like county mini-labs functioning as compact testing facilities for culture and sensitivity tests. These mini-labs exclusively focus on detecting the presence or absence of active pharmaceutical ingredients and other impurities in medicines.

Community-level misuse of drugs, especially antibiotics obtained without prescriptions, is concerning. Financial constraints often lead individuals to compromise prescribed treatment durations when visiting pharmacies. Corruption further undermines regulatory effectiveness through bribery and evasion. Extensive educational programs target healthcare professionals, including clinicians, lab technicians, and pharmacy staff, focusing on AMR and stewardship. Training courses prioritize capacity building using an approach focuses on the training of trainers. Specialized tools assist AMR champions in prescription assessments, promoting appropriate medication use. General awareness efforts also target the wider population.

## 5. Potential Ways Forward

Despite prevailing challenges, this study draws inspiration from successful global examples, notably the Bangladesh Rural Advancement Committee (BRAC)’s initiative in sustainable rural poultry development. This model, designed to empower women in village settings, holds promise for adaptation in Kenya. It encompasses thorough training in poultry rearing, vaccinations, financial support, and the provision of high-yielding variety cocks for crossbreeding, resulting in increased income, crop yield, safer hygiene practices, food security, and farmers being empowered towards self-reliance [6,52,53]. Importantly, this approach has the potential to reduce antibiotic dependence in poultry farming. 

Additionally, the study emphasizes the need for sustainable investments to boost research and development in novel medicines, diagnostics, vaccines, and interventions. Aligned with Kenya’s NAP, prioritizing affordable and user-friendly point-of-care diagnostics is crucial. To tailor these tools effectively to the Kenyan landscape, specific prerequisites must be addressed, including adapting diagnostic equipment to harsh conditions, championing low-tech, cost-effective, and easily maintainable equipment, ensuring accessible and affordable training, and embracing reverse innovation for practical application in low-resource environments [54].

The following section (Table 2) presents key findings and corresponding recommendations derived from the study’s analysis, offering a comprehensive overview of factors contributing to AMR, associated challenges, and proposed interventions to guide the way forward.

## 6. Limitations

This study has several limitations that should be considered when interpreting the findings. Firstly, the reliance on a systematic scoping review and single-expert interview may result in an incomplete understanding of the complex and diverse dynamics of AMR in Kenya. The translation of policy into practice and the ever-evolving nature of the country’s healthcare system suggests that the findings may not fully reflect the real-world implementation of AMR interventions. Lastly, while efforts were made to ensure rigor, it is important to acknowledge that the available data and sources may be subject to biases and limitations, which could potentially impact the accuracy of the study’s conclusions.

## 7. Materials and Methods

This study employed a two-fold approach to investigate Kenya’s efforts against antimicrobial resistance (AMR). Firstly, a comprehensive systematic scoping review integrated qualitative and quantitative data to identify key aspects of Kenya’s progress in addressing AMR, initially focusing on its drivers. The second methodology involved consulting an expert in the Kenyan healthcare system to validate the findings and support the conclusions of this study. Finally, the research explores effective strategies to curb AMR.

### 7.1. Scoping Review

The scoping review protocol followed the framework proposed by Arksey and O’Malley [55], guiding the study through five primary stages. The systematic scoping review also adhered to the PRISMA guidelines for Scoping Reviews (PRISMA-ScR), ensuring methodological rigor and transparency throughout the study [56].

#### 7.1.1. Identifying Relevant Studies

Extensive literature searches were conducted using various databases, with a focus on electronic databases to identify relevant articles. To ensure broad coverage and minimize potential gaps, a combination of databases was used: PubMed, Embase (Ovid) and Google Scholar. The following terms and Boolean operators were used: (AMR* or Antimicrobial Resistance* or Bacterial resistance* or Drug resistance*).MP. AND (Surveillance* or Diagnostics* or Epidemiology* or Stewardship* or Awareness*).mp. AND Kenya. MP. The results were refined to include only randomized controlled trials, reviews, or systematic reviews. Additionally, the Google Scholar database was used to screen for grey and unindexed literature [55]. 

Moreover, relevant government and policy reports, including texts and reports from reputable sources such as the WHO, GLASS, and the Ministry of Health (MOH), were also considered for inclusion in the study. This is particularly relevant because, in a number of low- and middle-income countries (LMICs), government and policy reports often serve as the primary and essential source of information on specific issues [57]. 

#### 7.1.2. Study Selection

The titles and abstracts of all the references were carefully examined to efficiently manage the screening process and eliminate duplicates. This initial screening identified potentially relevant articles for further full-text reviews, aligned with the research question. For government and policy reports without available abstracts, the author reviewed executive summaries or similar documents [58]. Articles selected for full-text review were individually assessed by the author to determine their alignment with the inclusion criteria. To ensure the identification of relevant studies, ‘reverse snowballing’ was used [59].

The following inclusion criteria were specified in this review: (1) article types encompassed cohort, RCT, case–control, case report, case series, reviews, editorials, and policy reports; (2) the publication date was restricted to articles published from 2001 onwards, justified by the need to capture contemporary trends and developments in AMR within the specified timeframe; (3) the review focused on analyzing a variable of interest related to AMR; and (4) language restrictions dictated that only originally English and Swahili language papers were considered for inclusion.

#### 7.1.3. Charting the Data

Data extracted from the full texts were systematically organized and recorded in a chart, including essential details such as the study title, author(s), location of the evaluation, method of evaluation, AMR indicators assessed, or specific aspects evaluated, and a summary of the key findings of the study. These dimensions were based on [23,60], with refinements incorporated during the review process.

#### 7.1.4. Summarizing and Reporting the Results

To illustrate the review process and document study exclusions with the associated reasons, a PRISMA diagram was utilized. This approach aims to identify commonalities and variations among the evaluations, highlighting significant data considerations.

### 7.2. Expert Opinion

As a complement to the analysis, a complimentary interview was conducted with a prominent expert in the Kenyan AMR realm to gather valuable insights and opinions on the study’s findings. The expert was sought after because of their extensive experience in addressing AMR across various public, private, and faith-based sectors. Their diverse expertise made them a valuable resource to provide insights from multiple perspectives on the challenges and approaches related to antimicrobial resistance. The interview was conducted online and lasted for approximately 30 min. Before the interview, the expert was provided with a detailed description of the study, ensuring a clear understanding of its objectives and scope. Written informed consent was obtained from the expert before proceeding with the interview.

## 8. Conclusions

In conclusion, the current status of AMR in Kenya presents a scenario of increasing resistance to common first-line drugs and a rise in life-threatening infections caused by drug-resistant pathogens. Despite this, Kenya has made commendable efforts to tackle AMR, through the development of an exceptional policy framework outshining numerous regional counterparts. The detailed NAP and establishment of collaborative partnerships involving both government and civil society organizations underscores the nation’s commitment. Notable initiatives, such as AMR stewardship committees, bolstered laboratory capabilities, and education and awareness programs have shown promising results. 

However, the situation becomes concerning when examining policy implementation and effectiveness. This study highlights the complex nature of Kenya’s AMR landscape, in which superb policies coexist with uncertain outcomes in practice. This is mirrored in this study’s exploration, which underscores a sense of fragmentation. However, the trajectory displays a promising dedication to addressing AMR. Sustaining progress and mitigating the impact of AMR will require the establishment of robust data gathering systems, exemplified by further studies such as this, that will serve as the cornerstone for informed policymaking. By systematically recording data on AMR trends and interventions, Kenya can empower evidence-based decision-making, driving impactful reforms, ultimately paving the way for the advancement of the health and well-being of the Kenyan populace.

## Figures and Tables

**Figure 1 antibiotics-13-00197-f001:**
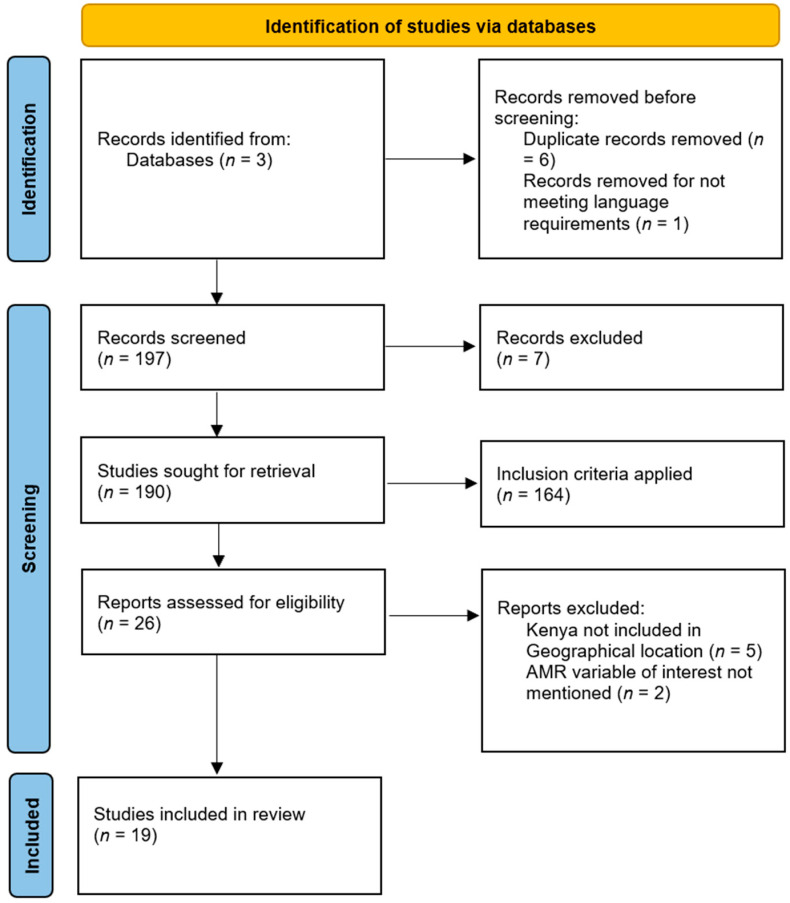
PRISMA-ScR flow diagram of the literature selection process [17].

**Figure 3 antibiotics-13-00197-f003:**
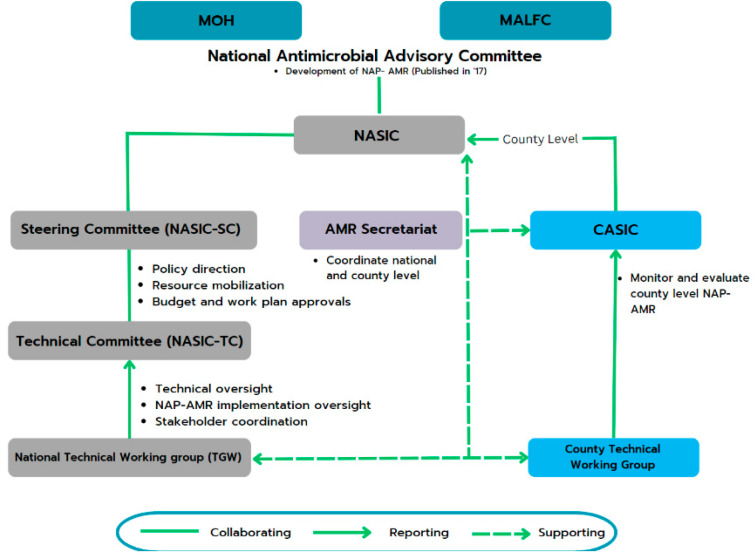
Kenya’s national- and county-level governance structure for implementing and monitoring the NAP on AMR [13]. Abbreviations: AMR, Antimicrobial Resistance; CASIC, County Antimicrobial Stewardship Interagency Committees; MALFC, Ministry of Agriculture, Livestock, Fisheries, and Cooperatives; MOH, Ministry Of Health; NAP, National Action Plan; NASIC, National Antimicrobial Stewardship Interagency Committee.

**Table 1 antibiotics-13-00197-t001:** Summarized below are the chosen publications, categorized by color, with **bold** highlighting non-peer-reviewed articles.

Citation	Setting	Study Design	Variable	Summarized Key Findings
**WHO, 2022 [5]**	Kenya	Government and policy reports	Review of Kenya’s NAP	Established functional national-level governance structures and advancing efforts to create county-level structures. Introducing antimicrobial consumption (AMC) surveillance.
**WHO, 2021 [7]**	Global	Government and policy reports	Review of Kenya’s GLASS progress	Reported to GLASS for two consecutive years and yet to submit AMR data for this year. Has 10 surveillance sites in the national system.
**Global Antibiotic Resistance Partnership—Kenya Working Group, 2011 [16]**	Kenya	Government and policy reports	AMR situational analysis	AMR concerns: Amplified by inadequate hospital infection control practice (IPC). Compounded by the presence of counterfeit drugs. Limited research and surveillance systems.
Joshi et al., 2021 [18]	Kenya among other LMICs	Quantitative situational analysis	Strengthening multi-sectoral coordination	Aimed to strengthen and implement stewardship in human health and in animal health sectors.
Moirongo et al., 2022 [19]	Kenya	Quantitative survey	Laboratory-based surveillance of AMR	Most labs lack bacterial culture testing services. External quality assessment program underutilized for cultures in Antimicrobial Susceptibility Testing (AST) facilities.
Omulo et al., 2017 [20]	Kenya	Qualitative study	AMR awareness in Kibera Informal Settlement	The majority of respondent’s lack understanding of antibiotics and proper usage. Healthcare workers have high trust among the populace. Opportunity for educational interventions.
Mbugua et al., 2020 [21]	Kenya	Cross-sectional Qualitative study	Perspectives of Hospital Managers on AMS	AMS implementation: Lacking due to the absence of core complementary health services. Health managers acknowledge the importance of AMS programs. Focus on implementing AMS in hospitals
Kimani et al., 2019 [22]	Kenya	Qualitative study	National One Health approach to AMR	Significant progress in addressing diverse health threats. Gaps exist: lacking extensive coordination for all One Health issues; focus needed on both zoonosis and AMR challenges.
Matee et al., 2023 [23]	Kenya, Tanzania, Uganda and Zambia	Mixed-methods Review	Performance in addressing AMR	Essential to strengthen National Integrated AMR Surveillance systems. Includes community settings and address AMR across human, animal, food, and environmental sectors. Crucial to establish a clear data-sharing protocol for AMR information in the region.
Iskandar et al., 2021 [8]	LMICS	Mixed methods Review	AMR Surveillance	LMICs show AMR surveillance progress. Tailored action plans are vital, reflecting individual country contexts. Alignment of regional, national, and international efforts is crucial.
Gulumbe et al., 2022 [9]	Africa	Mixed methods Review	Review on stewardship, surveillance, and diagnostics	Inadequate action in remote and primary healthcare settings. Lack of affordable diagnostic tools, stewardship programs, and surveillance. Increased funding, government legislation, enforcement, and civil society advocacy are needed.
Godman et al., 2022 [24]	Sub-Saharan Africa	Mixed-methods Review	Challenges and implications of tackling AMR	Limited personnel, expertise, capacity, and resources for NAP activities. Absence of focal points to lead NAPs. Conflicting demands and donor-related priorities.
Iskandar et al., 2020 [5]	LMICS	Literature review	Review of AMR from a “one Health” perspective	Limited resources, economic hardships, conflicts, epidemics, and political hurdles. Shift focus to individuals and governments. Prioritize education, training, research, and socio-ecological behavior change.
Othieno et al., 2020 [25]	Kenya	Literature review	AMR communication strategy	Enhancing NAP through Communication: Crucial to create effective communication strategies. Aims to drive behavioral changes among all involved in AMR efforts.
Kariuki et al., 2022 [26]	Sub-Saharan Africa	Literature review	Situational analysis of AMR rates and Surveillance.	Kenya stands out as a well-implemented NAP example in the region.
Acam et al., 2023 [27]	East Africa	Literature review	Antibiotic prescription patterns	Indicates inappropriate antimicrobial use, contributing to antimicrobial resistance. Study findings underscore the need for action to improve antimicrobial prescription practices.
Otieno et al., 2022 [28]	Sub-Saharan Africa	Literature review	Pharmacist-led AMS programs	Pharmacist-led interventions: Result in improved guideline adherence and reduced antimicrobial therapy and healthcare costs. Challenges include guideline absence, prescriber attitudes, lack of AMS teams, limited resources, and infrastructure.
Murila et al., 2022 [29]	Kenya	Retrospective review	Rational use of antibiotics at KNH	High incidence of irrational antibiotic prescribing in Kenyatta National Hospital critical care unit (CCU). Mainly attributed to incorrect choice and duration of use. Emphasizes proper antibiotic selection and duration for better management.
Kamere et al., 2022 [30]	Kenya among African nations	Scoping review	National Antimicrobial Stewardship Activities	Vital to align political commitment with investment in technical workforce capacity. Effective AMR addressing requires a holistic approach.

Abbreviations: AMC, Antimicrobial Consumption; AMR, Antimicrobial Resistance; AMS, Antimicrobial Stewardship; AST, Antimicrobial Susceptibility Testing; CCU, Critical Care Unit; ICP, Infection Control Practices; KNH, Kenyatta National Hospital; LMIC, Low-Middle-Income-Country; NAP, National Action Plan; GLASS, Global Antimicrobial Resistance and Use Surveillance System.

**Table 2 antibiotics-13-00197-t002:** Summarized findings and recommendations to address AMR.

Contributory Factors	Potential Issues	Proposed Interventions
Use of antibiotics in medical fields	- Unregulated sale and supply chain. - Lack of enforcement of current guidelines. - Consumer demand to eradicate symptoms. - Informal pharmacies. - Counterfeit medications.	- Stricter national rules and regulations. - Capacitate regulatory bodies. - Standard operating procedures. - Periodic audits. - Designate AMR champions to develop action plans at facilities. - Use ‘mini-labs’ to test for quality of medications.
Use of antibiotics in veterinary and agricultural fields	- Illegal veterinary outlets. - Therapeutic and prophylactic use in livestock production. - Cross-reservoir transmission.	- Promote a sustainable ‘BRAC’-like approach to rearing livestock. - Promote a ‘One Health’ approach across all AMR and IPC frameworks. - Formation of the Kenyan One Health office.
Healthcare divide	- Marginalized access to healthcare services. - Urban–rural divide for healthcare workers. - Complex and fragmented private sector. - Lack of vertical integration between public and private.	- Incentivize health workers to migrate to rural areas. - Standardize private sector AMR and IPC regulations. - Promote information sharing between public, private, and faith-based health facilities.
IPC, WASH and Immunization	- Implementation of county-wide IPC programs lacking. - Clean water and basic sanitation access lacking.	- Designate IPC and WASH champions across facilities. - Develop action plans to enhance systems. - Investment in basic sanitation and safe water access.
Amr Surveillance and Diagnostic capacity	- Questionable quality assurance. - Weak laboratory infrastructure. - Limited staff and training capacity. - Limited availability of consumables, diagnostics and reagents. - Lack of testing practices.	Promote decentralization by: - Exploring incremental decentralization with specific high-priority tests or services that can be managed with existing staff while building capacity. - Investing in R&D to foster Point-Of-Care testing, with diagnostics and reagents adapted to the specific environment. - Promote initiatives to build capacity, including focused training programs, recruitment drives, and collaborations with educational institutions to augment the pool of qualified staff. - Roll out community health workers and medical students. - Integrate AMR education into national training and provide educational grants for post-graduates. - Implement a standard operating procedure.
AMR awareness and communication	- AMR is a complex multifaceted issue. - Common misconceptions regarding AMR.	- National coordinating committee for communication and awareness - Training of trainer’s approach. - National awareness and education programs
AMS and Governance	- Top-down approach. - The majority of counties are not actively involved in NAP. - Environmental, food safety and production sectors are not involved in NAP. - Delay of the next iteration of NAP.	- Collaborative approach to include CSOs. - Promote information sharing between counties and national government. - Establish AMR and IPC committees at every facility.
Reliance on external funding	- Grants and funds may be short-term. - Lack of national budget for internal funding.	- Strengthen state capacity to steer structural transformation. - Build up an aid management system. - Government intervention for internal funding.

Abbreviations: AMR, Antimicrobial Resistance; BRAC, Bangladesh Rural Advancement Committee; CSO, Civil Society Organization; ICP, Infection Control Practices; LMIC, Low-Middle-Income-Country; NAP, National Action Plan; R&D, Research and Development; WASH, Water, Sanitation and Hygiene.

## Data Availability

All data stated in this review are available in the References cited.

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
