# Peer review of "The Fragmented Picture of Antimicrobial Resistance in Kenya: A Situational Analysis of Antimicrobial Consumption and the Imperative for Antimicrobial Stewardship"

_antibiotics, 2024, doi:10.3390/antibiotics13030197_

Round 1

Reviewer 1 Report

Comments and Suggestions for Authors

This is an interesting manuscript about AMR in the primary setting.

It is a well organised work written with proper use of English.
The limitations are addressed in the Discussion. 

Author Response

Dear Reviewer,

Subject: Response to Referee Comments on Manuscript ID [antibiotics-2797930] titled " The Fragmented Picture of Antimicrobial Resistance in Kenya: A situational analysis”.

I hope this letter finds you well. I would like to express my gratitude for the constructive feedback on our manuscript. We have carefully considered the comments provided and made revisions accordingly. Please find below our detailed response to your comments:

  1. Comment: This is an interesting manuscript about AMR in the primary setting.

Response: We appreciate the referee’s recognition of the significance of our work. We have strived to present a study addressing the AMR landscape in Kenya, including in the primary setting, and we are pleased that this interest has been conveyed effectively.

  1. Comment: It is a well-organized work written with proper use of English.

Response: Thank you for acknowledging the organization and language usage in our manuscript. We have ensured that the language used is precise and adheres to proper English conventions.

  1. Comment: The limitations are addressed in the Discussion.

Response: We are pleased that the referee recognized our efforts in addressing the limitations of our study in the Discussion section.

We sincerely hope that these revisions align with the expectations of the referee and the editorial team. Thank you once again for the invaluable feedback, and we look forward to the opportunity to contribute to MDPI.

On behalf of the research team,

Aarman Sohaili 

Reviewer 2 Report

Comments and Suggestions for Authors

Overall this paper is well-written and it summarizes the situation of AMR in Kenya. I agree that these factors are important not only in Kenya but also in other parts of the world, especially in some developing countries. 

I only have several minor comments for the authors:

1. It may be better for the authors to spell out the full name of abbreviations the first time you mentioned it in the manuscript. Although abbreviations such as IPC and WASH are well-known, not all readers understand the meaning of these terms.

2. In Table 2, the authors mentioned limited staff and training capacity, but at the same time proposed intervention is to decentralize laboratory capacities. With the limited staff number, it may be difficult to decentralize laboratory capacities to cope with regular ASTs for bacteria as mentioned in the manuscript. Are there any other interventions or measures that the authors can propose to tackle the problem of manpower shortage?

Author Response

Dear Reviewer,

Subject: Response to Referee Comments on Manuscript ID [antibiotics-2797930] titled " The Fragmented Picture of Antimicrobial Resistance in Kenya: A situational analysis”.

I hope this letter finds you well. I would like to express my gratitude for the constructive feedback on our manuscript. We have carefully considered the comments provided and made revisions accordingly. Please find below our detailed response to your comments:

  1. Comment: Abbreviation Spelt out

Response: We will ensure that the full name of abbreviations will be spelt out in the manuscript, including WASH, upon their first mention to ensure clarity for readers.

  1. Comment: Limited Staff and Training Capacity vs. Decentralization

Response: We acknowledge the referee's concern about the potential challenges of decentralizing laboratory capacities with limited staff. To address this, we will provide additional clarification in the manuscript, discussing potential alternative interventions or measures to mitigate the impact of manpower shortages (page 18).

We sincerely hope that these revisions align with the expectations of the referee and the editorial team. Thank you once again for the invaluable feedback, and we look forward to the opportunity to contribute to MDPI.

On behalf of the research team,

Aarman Sohaili 

Reviewer 3 Report

Comments and Suggestions for Authors

ANTIBIOTICS-MDPI

7.01.2024

Review report

"The Fragmented Picture of Antimicrobial Resistance in Kenya: A situational analysis"

Authors: Aarman Sohaili, Judith Asi and Pierre Thomas

Antimicrobial resistance and antibiotic stewardship are critical issues for medical practice worldwide.

The manuscript is a review based on literature research and a PRISMA diagram is provided.

Some aspects of antibiotic use in Kenya are described, but the approach omit the concrete data of community and hospital antibiotic resistance.  The deficiencies of keeping the rules for the rational use of antibiotics and the barriers in antibiotic stewardship are not specific for Kenya and are common to most of low-income countries, or unperforming heath policies.

I consider that the manuscript is not appropriate for the aim & scope of the Antibiotics Journal. 

I recommend resubmitting the manuscript to a journal of public health.

Comments on the Quality of English Language

No comment.

Author Response

Dear Reviewer,

Subject: Response to Referee Comments on Manuscript ID [antibiotics-2797930] titled " The Fragmented Picture of Antimicrobial Resistance in Kenya: A situational analysis”.

I hope this letter finds you well. We extend our gratitude to the referee for their thoughtful evaluation of our manuscript. We are grateful for the constructive feedback and the recognition of the significance of our work.

In response to the referee's comment regarding the omission of concrete data on community and hospital antibiotic resistance, we would like to provide clarification on our approach and intention for publication in the Antibiotics journal.

Our study's primary goal was not to offer a quantitative assessment of antibiotic resistance but to synthesize existing literature, creating a more comprehensive understanding of the antimicrobial resistance (AMR) landscape in Kenya. This approach is mentioned in our manuscript's intention and methodology. While we acknowledge the referee's concerns, we respectfully would like to state that, while a comprehensive and quantitative review of the burden of AMR infections in Kenya would be ideal to effectively tackle the Burden of Disease. However, the research team believes that the best estimates of AMR prevalence in Kenya must be contextualized using a thorough overview of the current policies in place at every level. This is what our paper tries to achieve by reviewing what is currently being undertaken at the regulatory level. Our hope is that researchers can leverage our paper to navigate the diverse responsibilities associated with AMR and ensure that their data contributes significantly to addressing the issue at hand.

Our study aligns with the Antibiotics journal's aim and scope, focusing on advancing research in AMR, misuse, and antimicrobial stewardship. Importantly, the journal welcomes both qualitative and quantitative research exploring the determinants of antimicrobial use and resistance, which our study comprehensively addresses. It is noteworthy that this specific concern was not raised by other referees or the editor.

Additionally, the referee rightly observed that the deficiencies in adhering to antibiotic use rules and barriers for antimicrobial stewardship (AMS) are not specific to Kenya but common in most low-income countries. In response, our paper offers recommendations based on expert assessments and real-life scenarios. Given the commonality of this issue across low-income countries, we believe our findings hold relevance and applicability beyond the Kenyan context.

We trust that these clarifications address the concerns raised by the referee. We remain open to any further suggestions or guidance to enhance the manuscript.

Thank you for your thoughtful consideration.

On behalf of the research team,

Aarman Sohaili 

Reviewer 4 Report

Comments and Suggestions for Authors

All reports concerning AMR from the regions with inadequate heath care, poor sanitary conditions, etc. are valuable and necessary sources of information. This is due to many ways of AMR spread all over the world.

The review presented fulfills requrements for this kind of report.

My concern is a single expert opnion, which metedologically and statistically is irrelevant.

Author Response

Dear Reviewer,

Subject: Response to Referee Comments on Manuscript ID [antibiotics-2797930] titled " The Fragmented Picture of Antimicrobial Resistance in Kenya: A situational analysis”.

I trust this letter finds you well. I extend my gratitude to the referee for their thoughtful evaluation of our manuscript. We are grateful for the constructive feedback and the recognition of the significance of our work.

We appreciate the referee's thoughtful consideration of our manuscript, and we welcome the opportunity to provide further clarification on the inclusion of a single expert opinion. While we wholeheartedly recognize the significance of methodological and statistical rigor in scientific endeavours, the inclusion of a solitary expert opinion in our review serves a specific and valuable purpose.

The expert opinion we incorporated into our manuscript represents a practitioner with profound immersion in the local healthcare landscape, offering nuanced insights that complement the quantitative data presented. Importantly, it is imperative to acknowledge the practical reality that the number of experts specializing in antimicrobial resistance (AMR) in Kenya is limited, a concern we explicitly addressed in our paper. We underscored the issue of ambiguity and the challenge of achieving local representation in AMR research in Kenya, which inherently affects the available pool of experts for consultation.

Our review stands on a foundation of an analysis of existing literature, and the inclusion of the expert opinion aligns with the intention to balance the quantitative emphasis with qualitative insights, thereby presenting a view of the AMR landscape in the context of our study.

In response to the concern that the inclusion of only one expert opinion may be perceived as statistically irrelevant, we would like to draw attention to a relevant study ([Reference: https://www.tandfonline.com/doi/full/10.1080/14787210.2021.1951705]) where authors addressed AMR in low- and middle-income countries. This study employed a similar approach, incorporating literature, relevant reports, policy documents, and the inclusion of a single expert opinion while maintaining statistical relevance. We believe this reference demonstrates a precedent for such methodology in the field.

It's worth noting that this particular concern was not raised by other referees or the editor, further emphasizing the nuanced nature of expert opinion inclusion in this context.

We hope this clarification addresses the concerns raised, and we remain open to any further discussions or suggestions to enhance the manuscript.

On behalf of the research team,

Aarman Sohaili 

Reviewer 5 Report

Comments and Suggestions for Authors

I suggest to authors:

1. review the size of the paragraphs, some paragraphs are too long, the ideal is paragraphs around 5 lines;

2. I suggest reviewing some references, for example: there is a reference that says the year of publication is 2033;

3. I suggest modifying some popular terms for more scientific language.

4. If possible, I suggest replacing older references with newer ones, for example: there is a reference from 2009.

Comments on the Quality of English Language

I suggest reviewing English. It has some popular words that are not applicable in scientific writing and small problems with agreement and coherence.

Author Response

Dear Reviewer,

Subject: Response to Referee Comments on Manuscript ID [antibiotics-2797930] titled " The Fragmented Picture of Antimicrobial Resistance in Kenya: A situational analysis”.

I trust this letter finds you well. I extend my gratitude to the referee for their thoughtful evaluation of our manuscript. We are grateful for the constructive feedback and appreciate the recognition of the significance of our work. In response to the referee’s suggestions, we have made the following adjustments:

  1. Comment: Paragraph length:

Response: We have reviewed the size of paragraphs, aiming for an optimal length of around five lines to enhance readability and flow.

  1. Comment: Reference Review:

Response: We have carefully examined all references, correcting any inaccuracies. Notably, we have rectified the reference that previously indicated the year of publication as 2033.

  1. Comment: Scientific Language:

Response: We have revisited the use of popular terms, substituting them with more precise scientific language where applicable. This modification is intended to enhance the scholarly tone and precision of our manuscript.

  1. Comment: Updated References:

Response: While we acknowledge the importance of utilizing recent literature, we would like to highlight a specific aspect of our study that necessitates the inclusion of older references. Our research delves into an issue where limited contemporary literature exists. The scarcity of recent publications on this specific topic underscores the enduring nature of the problem and the ongoing relevance of older sources. To enhance the comprehensiveness of our review, we have incorporated a mix of both older and more recent references. This approach ensures that our readers receive a well-rounded perspective on the historical development and current state of the issue at hand. Moreover, we have added supplementary references from recent years wherever possible to enrich our literature review.

We sincerely hope that this rationale resonates with the editorial board's expectations and standards for MDPI. Thank you for your thoughtful consideration, and we look forward to any further guidance or feedback.

On behalf of the research team,

Aarman Sohaili